# Atlas-Based Adaptive Hadamard-Encoded MR Spectroscopic Imaging at 3T

Huawei Liu [1], Adam W. Autry [1], Peder E. Z. Larson [1,2], Duan Xu [1,2] and Yan Li [1,*]

1    Department of Radiology and Biomedical Imaging, University of California San Francisco, San Francisco, CA 94107, USA; adam.autry@ucsf.edu (A.W.A.); peder.larson@ucsf.edu (P.E.Z.L.); duan.xu@ucsf.edu (D.X.)
2    UC Berkeley-UCSF Graduate Program in Bioengineering, University of California San Francisco, San Francisco, CA 94107, USA
*    Correspondence: yan.li@ucsf.edu; Tel.: +1-415-514-2104

**Abstract:** Background: This study aimed to develop a time-efficient method of acquiring simultaneous, dual-slice MR spectroscopic imaging (MRSI) for the evaluation of brain metabolism. Methods: Adaptive Hadamard-encoded pulses were developed and integrated with atlas-based automatic prescription. The excitation profiles were evaluated via simulation, phantom and volunteer experiments. The feasibility of γ-aminobutyric acid (GABA)-edited dual-slice MRSI was also assessed. Results: The signal between slices in the dual-band MRSI was less than 1% of the slice profiles. Data from a homemade phantom containing separate, interfacing compartments of creatine and acetate solutions demonstrated ~0.4% acetate signal contamination relative to the amplitude in the excited creatine compartment. The normalized signal-to-noise ratios from atlas-based acquisitions in volunteers were found to be comparable between dual-slice, Hadamard-encoded MRSI and 3D acquisitions. The mean and standard deviation of the coefficients of variation for NAA/Cho from the repeated volunteer scans were 8.2% ± 0.8% and 10.1% ± 3.7% in the top and bottom slices, respectively. GABA-edited, dual-slice MRSI demonstrated simultaneous detection of signals from GABA and coedited macromolecules (GABA+) from both superior grey and deep grey regions of volunteers. Conclusion: This study demonstrated a fully automated dual-slice MRSI acquisition using atlas-based automatic prescription and adaptive Hadamard-encoded pulses.

**Keywords:** MR spectroscopy; automatic prescription; Hadamard pulses; dual-slice; GABA



## 1. Introduction

Proton MR spectroscopy ($^1$H-MRS) is a powerful and non-invasive tool for investigating changes in brain metabolism associated with neurological and psychiatric diseases [1–6]. Although single-voxel spectroscopy is the most commonly used methodology, newer, fast 2D and 3D magnetic resonance spectroscopy imaging (MRSI) technologies [7,8], such as echo-planar spectroscopic imaging (EPSI) [9,10], non-Cartesian sampling [11–13] and parallel imaging techniques [14–16], are capable of evaluating the distribution of brain metabolites at a higher spatial resolution within clinically feasible acquisition times. Although 3D acquisitions enable much greater volumetric coverage, 2D MRSI has the advantage of providing a shorter acquisition time for clinical or time-limited research studies. The choice between 2D and 3D MRSI has largely depended on the regions of interest (ROIs), with larger or less focal ROIs generally favoring 3D acquisitions due to the inconvenience and added time of prescribing multiple 2D acquisitions.

Simultaneous multi-slice acquisitions using parallel imaging [17,18] and Hadamard encoding [19–27] techniques have previously been described as a means of overcoming this tradeoff. Parallel imaging techniques, such as sensitivity encoding (SENSE) [28–30], utilize the coil sensitivity map from multiple receive coil elements to decompose spatial information. Because the performance of these methods is adversely affected by coil

sensitivity profiles, inhomogeneous distribution of the geometry factor (g-factor) [17,28] and variable lipid suppression, Hadamard encoding has been widely employed for multi-volume spectroscopic imaging acquisitions [22,24,26,27]. It also improves the signal-to-noise ratio (SNR) by a factor of $\sqrt{N}$ relative to a single 2D MRSI acquisition, where N is the number of slices. A larger N provides higher SNR but at the cost of an increasing peak B1 amplitude when the pulses are superimposed. Consequently, pulse bandwidths must be reduced to maintain the maximum B1 value within hardware tolerances, resulting in possible chemical shift displacement errors (CSDEs).

Previous spectral studies have shown that many neurologic and psychiatric diseases have metabolic abnormalities within the motor and sensory cortex, cingulate cortex and subcortical structures, such as amyotrophic lateral sclerosis [31,32] and obsessive compulsive disorder [33]. The simultaneous detection of brain metabolism within these separate slice locations using automatic prescription would dramatically simplify acquisition procedures and benefit patient evaluation. Additionally, as the main neurotransmitters in adults, glutamate and γ-aminobutyric acid (GABA) have become important focuses of clinical research studies on disorders such as autism spectrum disorder [34], epilepsy [35,36] and schizophrenia [37,38]. The detection of GABA is most commonly achieved with an editing sequence at 3T, called GABA-edited, which takes advantage of J-coupling information to resolve the GABA peaks from the overlapping creatine signals in the difference spectrum of the two cycles, with editing pulses' on-and-off cycles [39]. The GABA-edited acquisition has been applied in single-voxel [39,40] or multiple-voxel modes [41–43].

The previously developed automatic atlas-based registration specified a prescription based upon pre-defined locations of the volume of interest (VOI) in the MNI152 standard space, and then the parameters that defined the position of the MRSI prescription relative to the subject's frame of reference were passed directly to the acquisition sequence. This method has been shown to achieve good inter-subject reproductivity in volunteer studies [44]. In this study, we developed and evaluated simultaneous dual-slice MRSI using Hadamard encoding with automatic prescription of the excited volume within superior and deep gray matter and assessed the feasibility of incorporating GABA-editing pulses.

## 2. Materials and Methods

### 2.1. Dual-Band Hadamard Pulse Simulation

Hadamard pulses were generated by vector superposition of single-band radio frequency (RF) pulses [19]. These pulses were generated using a Shinnar–Le Roux pulse design algorithm [45,46], with one optimized set being selected from Bloch-equation simulated results [47] according to the following criteria: in- and out-of-slice ripple amplitudes of 0.5% and 0.1%; pulse duration tp = 3.6 ms; 360 steps; 90° flip-angle; and 2.1 kHz bandwidth. The performance of Hadamard pulses with regard to slice selection was evaluated via simulation with varying slice separation, including 15, 20, 25, 30, 40 and 50 mm.

### 2.2. MRSI Sequence

Hadamard pulses were incorporated into the Point REsolved Spectroscopy (PRESS) MRSI sequence [48], which was also tailored to accept prescription parameters from a text file [44], by replacing the 90° RF pulse for localization. Compared to the original 90° RF pulse with 3.6 ms pulse duration and 2.3 kHz spectral bandwidth, Hadamard pulses had the same pulse duration and slightly smaller spectral bandwidth (2.1 kHz) but still exceeded the 933 Hz BW of 180° refocusing pulses. An overpress factor (factor increase of prescribed PRESS VOI) of 1.2 was applied to reduce CSDE [48]. Unwanted signals from the VOI periphery could be suppressed using outer volume suppression (OVS) pulses. Hadamard pulses can be generated adaptively based on the choice of interslice distance or the VOI exported from automatic prescription [44]. Spectral editing mode (BASING) [49] for GABA detection was enabled by placing 15 ms sinc-Gaussian editing pulses with 80 Hz bandwidths at 1.9 ppm and 7.5 ppm.

The sequence with adaptive dual-slice selection was evaluated in both phantoms and volunteers using a 32-channel array (Nova Medical, Wilmington, MA, USA) on a GE Discovery 750 3T scanner (GE Healthcare, Waukesha, WI, USA).

### 2.3. Phantom Imaging

The performance of dual-slice Hadamard-encoded MRSI was evaluated in two phantoms. Images of the excited volumes with 10 mm slice thickness were acquired from a standard GE MRS phantom (12.5 mM N-acetyl-aspartate, (NAA), 10 mM creatine (Cr), 3 mM choline (Cho), 12.5 mM glutamate, 7.5 mM myo-inositol and 5 mM lactate) with Hadamard pulses applied in the R/L direction. The distance, d, between the centers of the two slices was tested for 15, 20, 30 and 40 mm, respectively. No overpress factor was used to evaluate images of the excited volumes for slice localization. Metabolite crosstalk between the two slices was assessed using a homemade phantom composed of two interfacing compartments in a large container filled with purified water to reduce susceptibility: one compartment held 1% Cr solution, and the other 5% acetate (Ac) solution. Dual-slice Hadamard-encoded MRSI was acquired in the S/I direction to separately excite two slices covering Cr and Ac compartments, with 10 mm slice thickness, 1.25 s repetition time (TR), $10 \times 10$ mm$^2$ in-plane spatial resolution, $22 \times 22$ matrix size, flyback trajectory applied in R/L direction and 55 s total scan time. Slice profile errors were calculated as the ratio of Ac signal from the slice in the Cr compartment to Ac signal from the slice in the Ac compartment. Two single-slice 2D MRSI were also separately acquired from the Ac and Cr compartments and compared to dual-slice Hadamard-encoded MRSI for excitation profile and spectral quality.

The feasibility of detecting GABA was assessed from the SPECTRE phantom (Gold Standard Phantoms Inc., Sheffield, UK), which contains similar metabolites as the GE MRS phantom plus 2 mM GABA. Single-voxel, single-slice and dual-slice Hadamard-encoded GABA-edited MRSI were acquired with echo time (TE) = 68 ms and TR = 1.25 s.

### 2.4. In Vivo Imaging

Nine healthy volunteers (4 males, 5 females; age, $26 \pm 5$ years) who had no known history of neurological illness were recruited to this study following informed consent, as approved by the Institutional Review Board at UCSF. Six of the volunteers were studied for dual-slice Hadamard-encoded MRSI, with three of them receiving repeated scans after a brief break outside of the scanner room. The remaining three subjects were designated for testing the feasibility of incorporating GABA editing.

T1-weighted 3D inversion recovery spoiled gradient echo (IR-SPGR) images were acquired with inversion time (TI)/TE/TR = 450/2.5/6.6 ms, field of view (FOV) = $25 \times 25 \times 18$ cm$^3$ and spatial resolution = $0.5 \times 0.5 \times 1.5$ mm$^3$ for manual and automatic spectral prescription. The excitation region (VOI) covering superior motor/sensory cortex and deep gray matter structures and the OVS bands for suppressing subcutaneous lipid signals were predefined in MNI152 standard space [44]. Real-time transformation of the VOI and OVS to subject space was performed using custom-built software loaded on the scanner. For dual-slice Hadamard-encoded MRSI acquisitions, one additional saturation band was placed automatically between the two excited slices. Prior to all spectral acquisitions, high-order shimming was performed over the VOI.

Dual-slice Hadamard-encoded MRSI was acquired with two 10 mm slices placed at the top and bottom of the VOI, CHESS [50] water suppression, TE/TR = 144/1250 ms, matrix size = $22 \times 22$, nominal resolution = 1 cm$^3$, NEX = 4, flyback echo-planar trajectory applied in the R/L direction and total scan time ~4 min. For comparison, 3D MRSI was also acquired using the same atlas prescription template and similar acquisition parameters (e.g., nominal resolution = 1 cm$^3$), with flyback trajectory applied in the S/I direction, matrix size = $18 \times 18 \times 16$ and total acquisition time ~7 min [51]. An overpress factor of 1.2 was used to reduce CSDE [48].

Dual-slice Hadamard-encoded GABA-edited MRSI data were acquired with TE/TR = 68/1250 ms, matrix = 12 × 14, slice thickness = 10 mm and 15 × 15 mm$^2$ in-plane spatial resolution, resulting in a nominal voxel size of 4.5 cm$^3$.

### 2.5. Data Processing and Analysis

Processing was performed offline using the GE Orchestra package, custom-built scripts and SIVIC [52]. Spectral data acquired with flyback readout trajectories were first reordered to the corresponding rectilinear grid [10]. The 3D MRSI datasets were spatially shifted along the SI direction to match the center of the top or bottom slices of dual-slice Hadamard-encoded MRSI to compare differences between the two acquisitions. For dual-slice MRSI, spectra from the top or bottom slice were generated by summing and subtracting data from the two cycles. The difference spectra from GABA-edited data were created by subtracting the edited from non-edited cycles to measure the GABA signal. Spectral data were Fourier-transformed, phase- and frequency-corrected and coil-combined as previously described [51,53]. Peak heights and areas of Cho, Cr and NAA of the long TE MRSI datasets were computed automatically using in-house software [54], and the difference spectra of the GABA-edited datasets were quantified by LCModel [55] using a simulated basis set with sequence parameters and prior knowledge of chemical shifts and *J*-coupling [56] using NMRSCOPE-B from jMRUI [57,58].

Metabolite SNR was estimated by dividing the peak heights by the standard deviation of the noise from the peak-free region at the right end of the spectrum [54]. For comparison, the normalized SNR (nSNR) was calculated by dividing the SNR by spatial resolution and the square root of the total acquisition time. Intra-subject reproducibility of the ratios of peak heights was evaluated using the coefficients of variance (CoVs) from voxels with SNR >5 and linewidth <0.10 ppm. The mean and standard deviation (SD) were then calculated across subjects for top and bottom slices individually. For GABA editing, metabolite ratios included in the analysis were those with Cramer–Rao lower bounds (CRLB) < 20% from the editing-off spectra and <30% for GABA+ (GABA+ macromolecules) from difference spectra. Mean and SD of metabolite ratios (GABA+/NAA, Glx (glutamate + glutamine)/NAA) were calculated in the grey matter (GM) and white matter (WM) voxels that had at least 60% of GM (pGM = GM/(GM + WM) > 60%) or WM (pWM = WM/(GM + WM) > 60%), with GM, WM and cerebrospinal fluid segmented from $T_1$-weighted images [59]. The Pearson's linear correlation coefficients were calculated to determine the association between metabolite ratios and pGM.

## 3. Results

### 3.1. Dual-Slice Localization Using Hadamard Encoding

Examples of simulated excitation profiles for dual-band Hadamard pulses with 10 mm slice thickness and 30 mm slice distance are shown in Figure 1. Figure 2A,B display reconstructed image slices and calculated excitation profiles for different slice distances, with the average crosstalk signal between two slices < 1%. Figure 2C illustrates the results of spectral data from a homemade phantom. No visible Ac peak could be found in spectral data from the Cr slice, as Ac contamination in the excited slice from the Cr compartment represented 0.4% of the Ac compartment amplitude (below the noise floor). The ratio of the average SNR of Ac from dual-slice Hadamard-encoded MRSI to single-slice 2D MRSI was 1.32, slightly lower than the theoretical SNR gain factor of 1.41.

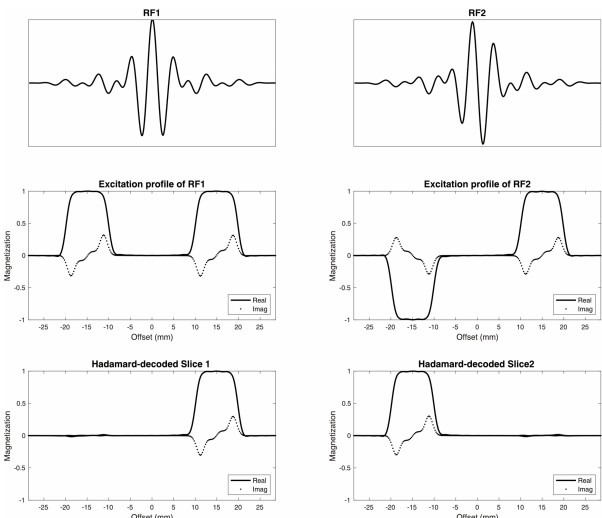

**Figure 1.** Dual-band Hadamard RF pulses. Dual-band Hadamard RF1 and RF2 pulses (**top**) with corresponding excitation profiles (**middle**) were designed for 10 mm slice thickness and 30 mm (**center-to-center**) slice distance (**bottom**).

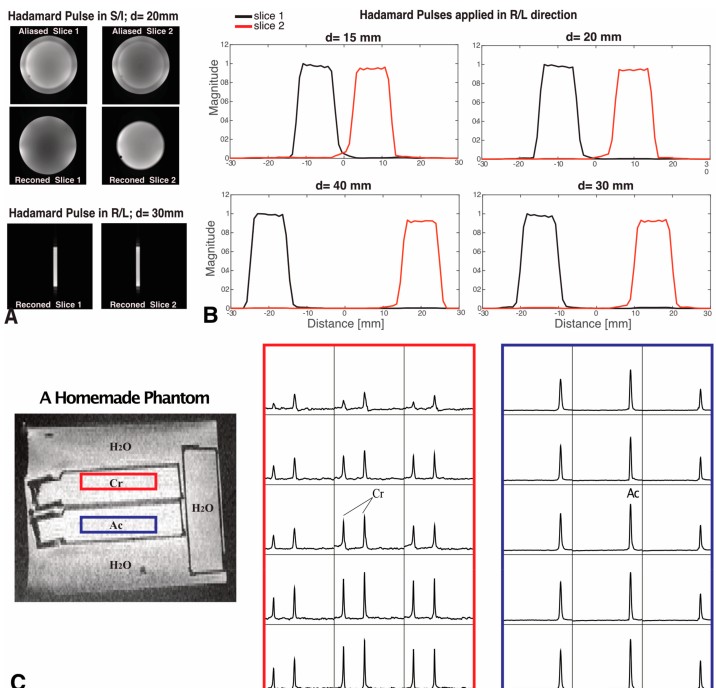

**Figure 2.** Hadamard pulse performance in phantoms. (**A**) Examples of slice images are shown when d = 20 mm (with Hadamard pulses applied in the S/I direction) and d = 30 mm (with Hadamard pulses applied in the R/L direction). Reconstructed (reconned) images were compared to the aliased images. (**B**) The slice profiles were calculated from the images acquired with Hadamard pulses applied in the R/L direction with 10 mm slice thickness and d = 15, 20, 30 and 40 mm, respectively. (**C**) Dual-slice Hadamard-encoded MRSI from a homemade phantom, where selective excitation is shown for compartments containing Cr (red box) and Ac (blue box) solutions. The slice selection gradient was applied in the S/I direction, and the center-to-center distance between slices was d = 26 mm. Spectra are displayed in the range of 1.2–4 ppm. No visible Ac peak could be found in spectral data from the Cr slice, demonstrating negligible crosstalk signal from the Hadamard pulse.

### 3.2. SNR and Reproducibility of Simultaneous Dual-Slice MRSI

Overall, in vivo dual-slice MRSI data were obtained with good quality. The mean $\pm$ SD *n*SNRs of NAA from dual-slice Hadamard-encoded MRSI and 3D MRSI were 14.3 $\pm$ 1.5 and 15.1 $\pm$ 2.1 in the top slices; and 8.4 $\pm$ 0.9 and 8.2 $\pm$ 1.6 in the bottom slices, respectively, and the linewidths of NAA were 6.1 $\pm$ 1.8 versus 5.1 $\pm$ 1.5 Hz (top slice) and 7.5 $\pm$ 2.8 versus 6.7 $\pm$ 3.0 Hz (bottom slice), respectively. The median Cho/NAA for dual-slice Hadamard-encoded MRSI and 3D MRSI were 0.49 versus 0.48 (top slice) and 0.67 versus 0.70 (bottom slice), respectively. Figure 3 illustrates automatically prescribed atlas-based spectra acquired from superior + deep GM regions of a volunteer using both dual-slice Hadamard-encoded MRSI (Figure 3A) and 3D MRSI (Figure 3B) for comparison; spectra from repeated dual-slice MRSI acquisitions are also shown (Figure 3C). The intersession CoVs of NAA and Cho from the repeat scans were 8.2% $\pm$ 0.8% (top slice, mean $\pm$ SD) and 10.1% $\pm$ 3.7% (bottom slice).

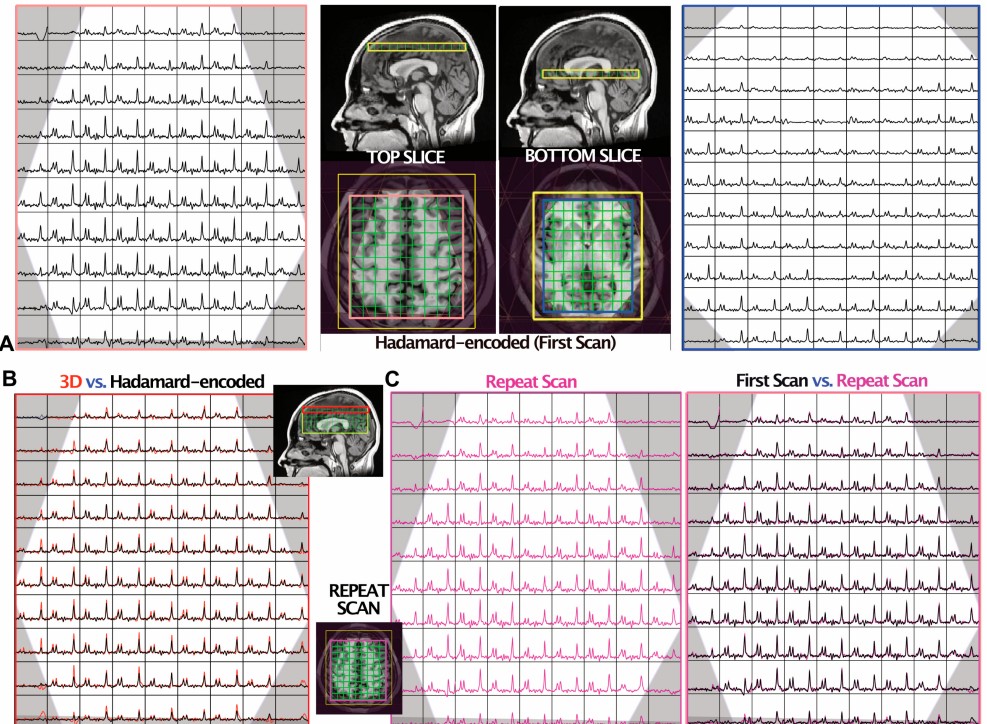

**Figure 3.** MRSI data from one volunteer. The OVS bands are shown in grey. (**A**) The spectra from two slides (top and bottom) were reconstructed from the dual-slice Hadamard-encoded MRSI acquisition. The median raw SNRs of NAA were 29 and 20 in the top and bottom slices, respectively. (**B**) Comparison of 3D MRSI (red) versus dual-slice Hadamard-encoded MRSI (black; (**A**) left) from the top slice, with the median raw SNR of NAA being 41 and 29, respectively. The 3D MRSI dataset was spatially shifted to match the top slice during post-processing, and the vertical scales of spectra were adjusted to equalize the noise level of the two spectra datasets. The mean $\pm$ SD linewidths of NAA were 6.5 $\pm$ 1.6 and 5.1 $\pm$ 1.8 Hz for the Hadamard-encoded MRSI and 3D MRSI, respectively. (**C**) A repeat scan was obtained to evaluate the intra-subject reproducibility of dual-slice Hadamard-encoded MRSI. The spectra from the repeat acquisition are shown on the left (magenta) and overlaid on the first acquisition (black; (**A**) left), which are displayed with the same intensity range. The CoV of NAA/Cho within this slice was 7.4%.

### 3.3. GABA-Edited Dual-Slice MRSI

The spectral editing in the GABA phantom showed that GABA/NAA was 0.11 in SVS, and the mean $\pm$ SD of GABA/NAA was 0.12 $\pm$ 0.02 for single-slice 2D MRSI and 0.12 $\pm$ 0.02/0.13 $\pm$ 0.02 for top/bottom slices of dual-slice Hadamard-encoded MRSI.

Metabolite ratios (GABA+/NAA, Glx/NAA) and CRLBs of GABA+, Glx and NAA in the GM and WM regions from the three volunteers are given in Table 1. Examples of spectral data from one volunteer are shown in Figure 4. The higher percent of GM tissue composition within the spectral voxel was associated with high levels of GABA+/NAA (correlation coefficient, $r = 0.26$, $p = 0.0002$) and Glx/NAA ($r = 0.393$, $p < 0.0001$).

**Table 1.** Metabolite ratios (mean $\pm$ SD) and CRLBs of metabolites quantified by LCModel in the GM and WM of 3 healthy volunteers.

| Location | | GABA+/NAA | GABA+/Cr | GABA+ CRLB | NAA CRLB | Cr CRLB |
|---|---|---|---|---|---|---|
| Top slice | GM | 0.16 ± 0.02 | 0.21 ± 0.02 | 16 ± 2 | 2 ± 0.3 | 3 ± 0.4 |
| | WM | 0.14 ± 0.02 | 0.22 ± 0.03 | 16 ± 2 | 2 ± 0.2 | 3 ± 0.2 |
| Bottom slice | GM | 0.18 ± 0.02 | 0.21 ± 0.04 | 13 ± 5 | 2 ± 0.2 | 3 ± 0.4 |
| | WM | 0.16 ± 0.02 | 0.23 ± 0.04 | 15 ± 2 | 2 ± 0.3 | 3 ± 0.4 |

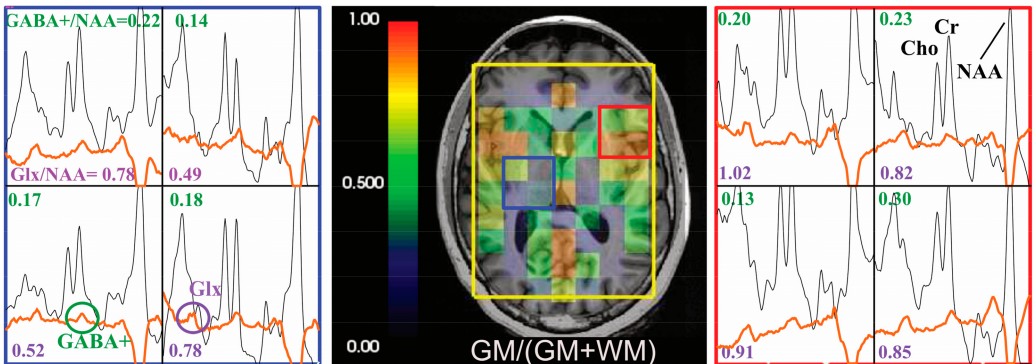

**Figure 4.** Hadamard pulse application to GABA editing. Examples of the difference spectra (before quantification using LCModel, with baseline, orange) and non-edited spectra (black) are presented for a volunteer. The corresponding GABA+/NAA (values in green) and Glx/NAA (values in purple) are shown in the spectra array. The GM tissue composition at the spectra resolution was overlaid on T1-weighted images.

## 4. Discussion

A fully automated dual-slice MRSI acquisition using atlas-based automatic prescription and adaptive Hadamard-encoded pulses was implemented with the option for GABA editing. Both phantom and volunteer results demonstrated accurate slice localization with negligible crosstalk between slices, reproducible metabolite ratios with good SNRs and the feasibility of detecting GABA/GABA+.

Though dual-slice Hadamard pulses from previous studies often displayed fixed slice thickness and gap distances [22,24], this work further developed a pulse sequence that generated Hadamard pulses automatically and adaptively based on the size of the VOI, slice thickness and the orientation of excitation. Defining a spectral prescription that both covers ROIs and suppresses regions with high lipid signals is time-consuming and creates user-dependent variabilities. The automated prescription method [44,60–64] has proved capable of overcoming these challenges and improving reliability with dice overlap ratios of 92–95% for SVS [53] and 97.8% for multi-voxel acquisitions [44]. By predefining templates in MNI152 standard space, this technique enables comparative evaluation of metabolism from disparate anatomic locations using dual-slice or two single-voxel acquisitions.

Imaging excitation profiles of the adaptive Hadamard pulses in the phantom showed accurate localization, uniform excitation and minimal crosstalk. The dual-slice Hadamard-encoded MRSI acquired from the homemade phantom showed negligible crosstalk (<1%) between the two slices, indicating narrow transition bands in the pulse profile and strong

pulse performance for slice-selective excitation. Compared to the single-slice 2D MRSI acquisition, MRSI acquired with dual-band Hadamard pulses displayed an Ac SNR gain of 32%. This gain was slightly less than the theoretical value of 41%, perhaps due to field inhomogeneity and the reduced spectral width of Hadamard pulses (2.1 kHz) relative to single-slice RF pulses (2.3 kHz).

As shown in Figure 3, spectral slices from dual-slice Hadamard-encoded MRSI were matched to the same spatial locations of 3D MRSI, demonstrating the accuracy of the proposed method for dual-slice excitation in in vivo acquisitions. Dual-slice Hadamard-encoded MRSI provided similar nSNR relative to 3D MRSI, thereby offering more flexibility in designing imaging protocols to balance scan time, SNR and voxel size. CoVs calculated on a voxel-by-voxel basis were comparable to previous reports [44,61]; and, as expected, the bottom slice displayed larger CoVs than the top slice due to lower SNR and broader linewidths in deep GM structures.

Both phantom and volunteer data demonstrated the feasibility of combining GABA editing with dual-slice Hadamard encoding MRSI. In the GABA phantom, the ratios of GABA/NAA were similar among SVS, single-slice 2D MRSI and Hadamard encoding methods. For in vivo data, the placement of editing pulses symmetric to the water resonance yielded co-editing of MM and GABA+, commonly used instead of GABA. Similar to previous MRSI studies [65], we also found that higher levels of GABA+/NAA and Glx/NAA ratios were statistically significantly associated with a higher percent of GM tissue composition, as well as higher mean GABA+/NAA ratios in the cingulate cortex and subcortical structures (bottom slice) compared to the motor and sensory cortex (top slice) (Table 1); ratios of GABA+/tCr were consistent with another study [43]. Although we only acquired three datasets in healthy controls, additional measurements in the future will help improve the robustness of the evaluation of anatomic GABA mapping.

Because the present study evaluated SLR pulses with Hadamard encoding in the standard PRESS sequence, there were some notable limitations. Importantly, localization sequences like sLASER [66], which utilize adiabatic pulses that are insensitive to B1 inhomogeneity, can produce more uniform excitation profiles compared to PRESS. The Hadamard encoding module that we developed could easily be integrated into a sLASER sequence by replacing the 90° excitation pulse, along with the automated prescription tool. Although Hadamard pulses that simultaneously encode two slices may not provide sufficient anatomical coverage in some applications, increasing the number of slices without sacrificing RF pulse bandwidth and maximum power considerations will be challenging. Additionally, Hadamard-encoded acquisitions are sensitive to motion [67], especially for GABA editing that requires much longer acquisition times to accommodate four spectral cycles and additional subtractions. Motion correction [42,65] that was not considered in this study might be needed for patient studies.

## 5. Conclusions

In conclusion, we have developed and implemented a simultaneous dual-slice MRSI method that is easy to use and provides high-quality reproducible spectral data, which may improve the clinical translation of 2D MRSI for the study of neurological and psychiatric diseases.

**Author Contributions:** Conceptualization, P.E.Z.L., D.X. and Y.L.; collection of data, H.L. and A.W.A.; data analysis, H.L., A.W.A. and Y.L.; writing—original draft preparation, H.L., A.W.A. and Y.L.; writing—review and editing, all authors. All authors have read and agreed to the published version of the manuscript.

**Funding:** This research was funded by NIH R21 HD092660, R01 CA273028 and R01 CA262630.

**Institutional Review Board Statement:** The study was conducted in accordance with the Declaration of Helsinki and approved by the Institutional Review Board of University of California San Francisco (IRB 10-01708 and date of approval 21 January 2020).

**Informed Consent Statement:** Informed consent was obtained from all subjects involved in the study.

**Data Availability Statement:** The data presented in this study are available on request from the corresponding author.

**Conflicts of Interest:** The authors declare no conflict of interest.

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
