# Peer review of "Atlas-Based Adaptive Hadamard-Encoded MR Spectroscopic Imaging at 3T"

_tomography, doi:10.3390/tomography9050127_

Round 1
Reviewer 1 Report
The authors proposed a new method for simultaneous, dual-slice MR spectroscopic imaging (MRSI) using Hadamard encoding with automated prescription, and assessed its performance for GABA-edited dual-slice MRSI in volunteers. This is a well conducted study. However, the following points require clarification and/or correction.
1. Introduction, page 1, line 32-34: The references for the following sentence should be corrected.
‘While single-voxel spectroscopy [7] is the most commonly used methodology, newer fast 2D and 3D magnetic resonance spectroscopy imaging (MRSI) technologies [8,9], such as echo-planar spectroscopic imaging (EPSI) [10,11], non-Cartesian sampling [12-14] and parallel imaging techniques [15-19]’
Although reference number 7 is a single voxel MRS paper, it is mainly a comparison of STEAM and PRESS. Adding a reference from 1989 for single-voxel spectroscopy might be unnecessary. Reference number 9 is the consensus paper for advanced MRSI neuroimaging, and it could be replaced by a paper on fast MRSI that is more relevant (such as Nelson et al., DOI: 10.1016/j.jmr.2013.02.003). References 17 and 18 should be removed, they are not parallel MRSI papers, but the original GRAPPA and SMASH papers. Reference number 19 is an old conference proceeding, which might possibly be replaced by later publications.
2. Introduction, page 2, lines 60-62: This sentence is incomplete. ‘Additionally, as the main neurotransmitters in adults, glutamate and gamma-aminobutyric acid (GABA) have become important focuses of clinical research studies, such as autism spectrum disorder [37], epilepsy [38,39], and schizophrenia [40,41].’
3. The sentences explaining the automated MRSI prescription and its previous successful application in the volunteers at the end of the last paragraph of the Introduction do not seem in place. It would be better to introduce this concept earlier in the Introduction and reserve the last sentence of the introduction to possibly state that the sequence was evaluated in both phantoms and volunteers at 3T.
4. Materials and Methods, page 2, line 90: What is the word ‘tailed’? Is it ‘tailored’?
5. Materials and Methods, page 3, line 95-97: These sentences describe a general concept, please replace them with what was actually conducted in this study. ‘Unwanted signals from the VOI periphery could be suppressed using outer volume suppression (OVS) pulses. Hadamard pulses can be generated adaptively based on the choice of interslice distance or the VOI exported from automatic prescription [47].’
6. Materials and Methods, page 3, 120-121: Please explain what was compared between the two single-slice 2D MRSI acquired from the Ac and Cr compartments and the dual-slice Hadamard-encoded MRSI.
7. Materials and Methods, page 3, 131: TI (inversion time) should be inversion time (TI).
8. Materials and Methods, page 3, 141: CHESS requires a reference.
9. Materials and Methods, page 3, 146: Please write out what CSRA is.
10. Materials and Methods, page 4, 155: This sentence is confusing, ‘For dual-slice MRSI, each slice (top or bottom) was generated by summing and subtracting the two cycles.’ It reads like top or bottom slices were obtained by summing or subtracting. Please rewrite it in a more clear way.
11. Materials and Methods, page 4, 164: What was jMRUI used? It is a software package, and it does not provide spectral priors.
12. Figure 2: Please add ppm values at x axis.
13. Figure 3: A repeat scan is first mentioned at figure 3. Please add it to the materials and methods.
14. Discussion, page 8, 264: What is dice overlap ratios? It has not been reported in this paper and was just introduced in the Discussion.
Reviewer 2 Report
The authors present their finding using their Hadamard-encoded, dual-slice, MR spectroscopic imaging sequence. The novelty of this paper is the calculation of the needed RF pulse profile for the Hadamard encoding based on the automatic positioning of the two slices guided by an atlas, a method previously developed in the same group. Other dual-slice methods lack this flexibility and above all, the automatization of the procedure presented in this paper to make it easier to use in a clinical setting. The methods and results are clearly described with the right amount of detail. In general, the conclusion is clear and the discussion describing the methods’ limitations is fair. I would like to recommend this work for publication after minor revision.
Major comment.
The differences found between GM and WM in table 1 do not seem to be statistically significant with 3 volunteers, correct? Therefore, line 290-295 in the discussion section should be removed or rewritten as you cannot claim higher ratios without any statistical significance.
Minor comments
- I really would appreciate some explanation about the results shown in table 1. It is my understanding that the CRLB values for GABA+, NAA and Cr are mentioned and not the concentrations of these metabolites. I am used to reporting the CRLB in mM, but guess the values mentioned in table 1 are the errors in percentage? It would make sense as the errors scale with the intensities found for NAA and Cr. Please comment and clarify in the table.
- The difference in the spectra for the 3D vs Hadamard-encoded are largest on the edge of the FOV (figure 3B, first two columns). Could the authors please comment on the reason behind this? Is it related to the point spread function of the 3D-CSI vs the slice-selection method of the Hadamard-encoded 2D-CSI?
- Although the size of the PRESS selection box can be derived from figure 3, the size of this PRESS box is not defined in the text. Please add this to the method section.
- Is the voxel size of the 3D-PRESS/CSI the same as for the 2D Hadamard encoded sequence? So, they differ slightly in FOV? Please comment and clarify in the method section.
- Figure 3, please add the location of the saturation bands to the figure. Either in the legend or by arrow/text in the figure.
- Figure 4 is difficult to understand unless the reader is an expert on GABA/Glx edited MR spectroscopy.
o Please add to the legend of figure 4 that the orange, bold line is the difference spectrum.
o Please label Glx, GABA, Cr and NAA in the figure. It would also help to indicate which of the signals are relevant metabolite signals and which are artefacts of the editing method.
o It is not clear what the numbers mean in the inset of the spectra of the figure. Those numbers do not correspond to the values of table 1? (example, GABA+/NAA=0.22 0.14 (orange) and Glx/NAA=0.78.
